# Distribution of OGTT-Related Variables in Patients with Cystic Fibrosis from Puberty to Adulthood: An Italian Multicenter Study

**DOI:** 10.3390/jpm13030469

**Published:** 2023-03-03

**Authors:** Andrea Foppiani, Fabiana Ciciriello, Arianna Bisogno, Silvia Bricchi, Carla Colombo, Federico Alghisi, Vincenzina Lucidi, Maria Ausilia Catena, Mariacristina Lucanto, Andrea Mari, Giorgio Bedogni, Alberto Battezzati

**Affiliations:** 1ICANS-DIS, Department of Food Environmental and Nutritional Sciences, University of Milan, 20133 Milan, Italy; 2Cystic Fibrosis Unit, Department of Pediatric Subspecialties, Bambino Gesù Children’s Hospital, IRCCS, 00165 Rome, Italy; 3Pediatric Cystic Fibrosis Center, Fondazione IRCCS Ca’ Granda Ospedale Maggiore Policlinico, Via Com-menda 9, 20122 Milano, Italy; 4Department of Pathophysiology and Transplantation, Università degli Studi di Milano, 20122 Milano, Italy; 5Cystic Fibrosis Hub Center, Azienda Ospedaliera Universitaria Policlinico G. Martino, 98125 Messina, Italy; 6Institute of Neuroscience, National Research Council, 35127 Padova, Italy; 7Department of Medical and Surgical Sciences, Alma Mater Studiorum-University of Bologna, 40126 Bologna, Italy; 8Internal Medicine Unit Addressed to Frailty and Aging, Department of Primary Health Care, S. Maria delle Croci Hospital, AUSL Romagna, 48121 Ravenna, Italy; 9Clinical Nutrition Unit, Department of Endocrine and Metabolic Medicine, IRCCS Istituto Auxologico Italiano, 20100 Milan, Italy

**Keywords:** cystic fibrosis, glucose tolerance, insulin secretion, oral glucose tolerance test

## Abstract

Background: Insulin secretion and glucose tolerance is annually assessed in patients with cystic fibrosis (PwCF) through oral glucose tolerance tests (OGTTs) as a screening measure for cystic fibrosis-related diabetes. We aimed to describe the distribution and provide reference quartiles of OGTT-related variables in the Italian cystic fibrosis population. Methods: Cross-sectional study of PwCF receiving care in three Italian cystic fibrosis centers of excellence, from 2016 to 2020. We performed a modified 2-h OGTT protocol (1.75 g/kg, maximum 75 g), sampling at baseline and at 30-min intervals, analyzing plasma glucose, serum insulin, and C-peptide. The modified OGTT allowed for the modeling of β cell function. For all variables, multivariable quantile regression was performed to estimate the median, the 25th, and 75th percentiles, with age, sex, and pancreatic insufficiency as predictors. Results: We have quantified the deterioration of glucose tolerance and insulin secretion with age according to sex and pancreatic insufficiency, highlighting a deviation from linearity both for patients <10 years and >35 years of age. Conclusions: References of OGTT variables for PwCF provide a necessary tool to not only identify patients at risk for CFRD or other cystic fibrosis-related complications, but also to evaluate the effects of promising pharmacological therapies.

## 1. Introduction

Cystic fibrosis-related diabetes (CFRD) is the most common comorbidity in patients with cystic fibrosis (PwCF), with a reported prevalence increasing with age, and affecting approximately 40% of adult PwCF [1,2]. Abnormalities of insulin secretion, compromised nutritional status, and more severe lung inflammation are all associated in patients with CFRD [3]. These conditions accelerate the decline in lung function, ultimately leading to lower survival.

In recent years, the introduction of modulator therapies directly targeting the underlying defect of cystic fibrosis changed the natural history of the disease, improving both the nutritional status and pulmonary function of PwCF [4]. While CFTR modulator therapy of any kind does not constitute a cure for CFRD to this day, only limited data are available on long-term effects and early (pre-puberal) administration. Of available CFTR modulator therapies, the ivacaftor monotherapy showed the most promising results [5,6], while the lumacaftor/ivacaftor studies targeting delta F508 homozygous patients showed no significant effects [7,8,9,10,11]. Even the new elexacaftor/tezacaftor/ivacaftor therapy is showing, at best, modest improvement on glucose tolerance in preliminary studies [12,13,14,15,16,17]. Initiating modulator therapies at a young age has the potential to preserve β cells functionality, but the detection of these seemingly small effects needs reference data to compare longitudinal data to the natural history of glucose tolerance and insulin secretory parameters.

CFRD may remain clinically silent for years, with insulin secretion defects beginning earlier than glucose intolerance [18]. The presence of this defect in early ages has severe clinical implication: they are related to the future worsening of glucose tolerance and CFRD [19,20]; they are associated with lung disease in young PwCF with mild to normal pulmonary function [21] independently from hyperglycemia [22]; and insulin is also an important anabolic hormone, and the catabolic effect of insulin insufficiency has important implication on growth [23,24], and is specifically associated with reduced adult height [25].

PwCF undergo annual screening for CFRD with an oral glucose tolerance test (OGTT), starting at ten or even six years of age [26]. Considering that standard OGTT cannot detect insulin secretion defects, we implemented a modified OGTT protocol in three separate and geographically distributed cystic fibrosis Italian centers, expanding the previously studied Milan cohort of PwCF [27]. The modified OGTT protocol included insulin and C-peptide measurement in addition to glucose, allowing for the modeling of parameters describing β cell function, insulin secretion, insulin clearance, and OGTT insulin sensitivity [28,29].

The aim of this study is to describe cross-sectionally at the Italian level the progression of OGTT parameters, β cell function, insulin clearance, and insulin sensitivity up to adulthood, while also providing national references for these parameters.

## 2. Materials and Methods

### 2.1. Study Design, Setting, and Participants

This was a cross-sectional study of the Italian cystic fibrosis population that consecutively enrolled patients between 2016 and 2020.

Patients were recruited from three Italian cystic fibrosis centers of excellence, selected to represent the northern, central, and southern geographical areas of Italy: the Cystic Fibrosis Centre of the Fondazione IRCCS Ca’ Granda Ospedale Maggiore Policlinico, Milan, Italy; the Cystic Fibrosis Unit of the Bambino Gesù Children’s Hospital, Rome, Italy; and the Cystic Fibrosis Referral Center of the University Hospital G. Martino, Messina, Italy.

To be eligible, patients had to be clinically stable in the previous 3 weeks (absence of major clinical events including pulmonary exacerbations, no change in their habitual treatment regimen including introduction of antibiotics or steroids). Exclusion criteria were diagnosis of CFRD, or treatment with insulin or oral hypoglycemic agents in the previous 6 months.

The study was conducted according to the guidelines of the Declaration of Helsinki and approved by the Ethics Committee of the University of Milan (protocol code 53/19, 26 November 2019). Informed consent was obtained from all subjects involved in the study.

### 2.2. Variables and Measurements

Outcomes for the main analysis were:OGTT parameters: glucose, insulin, and C-peptide (sampled before and at 30, 60, 90, and 120 min of the OGTT), and their area under the curve (AUC).β cell function: β cell glucose sensitivity, basal insulin secretion, insulin secretion at a fixed glucose concentration, total insulin secretion.insulin clearance: basal and OGTT insulin clearance.insulin sensitivity: quantitative insulin-sensitivity check index (QUICKI, for basal insulin sensitivity), and a 2-h oral glucose insulin sensitivity index (2-h OGIS for OGTT insulin sensitivity).

Predictors of the outcomes were age (continuous; in years), sex (categorical; 0 = Female, 1 = Male), and pancreatic sufficiency status (categorical; 0 = pancreatic sufficient, 1 = pancreatic insufficient).

#### 2.2.1. CFTR Gene Mutation, Clinical Evaluation, Anthropometric and Pulmonary Assessment

CFTR mutations were classified by epidemiological prevalence (F508del homozygous, F508del heterozygous, other) and combining the type of CFTR defect with clinical severity (class I to VI, with decreasing level of severity) [30].

All patients were evaluated before enrollment in the study. The clinician informed the patient of the study procedures and collected informed consent. Clinical records of the following variables were collected: pancreatic insufficiency, intermittent and chronic infections, history of lung transplant, and CFTR modulator therapy.

Anthropometric assessment consisted of measuring weight and height following standard procedures [31]. Body mass index (BMI) was calculated as weight in kilograms divided by the square of height in meters. For patients < 20 years, standard deviation scores were calculated based on the Centers for Disease Control and Prevention growth charts for weight, height, and BMI [32]. Patients were classified according to BMI in underweight (BMI < 18.5 kg/m^2^), normal weight (BMI 18.5–24.9 kg/m^2^), overweight (BMI 25–29.9 kg/m^2^), or obese (BMI ≥ 30 kg/m^2^), and for patients ≤20 years old according to BMI z-score [33]. Patients were also classified following the Cystic Fibrosis Foundation recommendations: for women BMI ≥ 22, for men BMI ≥ 23, and for people younger than 21 years old ≥ 50th percentile [34].

Spirometry was performed according to the American Thoracic Society and European Respiratory Society guidelines [35]. The forced vital capacity (FVC) and forced expiratory volume in 1 second (FEV1) were expressed as percentage of the reference values [36].

#### 2.2.2. Oral Glucose Tolerance Test and Laboratory Exams

All subjects received a 2-h OGTT (1.75 g/kg, maximum 75 g) sampling at baseline, and at 30-min intervals, the subjects had determinations of plasma glucose, serum insulin, and C-peptide concentrations. Based on plasma glucose concentrations, patients were classified in glucose tolerance categories according to [18].

On the same day, C-reactive protein, and glycated haemoglobin (HbA1C) were also measured.

Plasma glucose was measured on fluoride plasma samples (Gluco-quant; Roche/Hitachi analyser; Roche Diagnostics), and the other analytes were measured by commercial assays (ECLIA-Cobas C6000; Roche Diagnostics).

Based on the plasma glucose concentrations, patients were assigned to one of the following categories of glucose tolerance [18]: normal glucose tolerance, normal glucose tolerance with impaired fasting glucose, indeterminate glucose tolerance, impaired glucose tolerance, cystic fibrosis-related diabetes without fasting hyperglycaemia, and cystic fibrosis-related diabetes with fasting hyperglycaemia.

#### 2.2.3. Modeling of β Cell Function and Other OGTT-Derived Indices

Beta-cell function was assessed by modeling from OGTT glucose and C-peptide, as previously described [27,29,37], using a model that describes the relationship between insulin secretion and glucose concentration. The model expresses insulin secretion as the sum of two components. The first component represents the dependence of insulin secretion on absolute glucose concentration at any time point during the OGTT through a dose-response function relating the two variables. Characteristic parameter of the dose-response is the mean slope over the observed glucose range, denoted as β-cell glucose sensitivity. The dose-response is modulated by a potentiation factor, which accounts for the fact that during an acute stimulation, insulin secretion is higher in the descending phase of hyperglycemia than in the ascending phase at the same glucose concentration. As such, the potentiation factor encompasses several potentiating mechanisms, including prolonged exposure to hyperglycemia, non-glucose substrates, gastrointestinal hormones, and neural modulation. It is set to be a positive function of time and is constrained to average unity during the experiment. In normal subjects, the potentiation factor typically increases from baseline to the end of a 2-h OGTT [29]. To quantify this excursion, we calculated the ratio between the 2-h and the baseline value. This ratio is denoted as potentiation ratio. The second insulin secretion component represents the dependence of insulin secretion on the rate of change of glucose concentration. This component is termed the derivative component and is determined by a single parameter, denoted as rate sensitivity. Rate sensitivity is related to early insulin release [29].

The β cell function parameters derived from the model were β cell glucose sensitivity, i.e., the slope of the relationship between insulin secretion and glucose concentration, and basal and total OGTT insulin secretion.

Basal insulin clearance was calculated as the ratio between basal insulin secretion and concentration, and OGTT insulin clearance as the ratio of total insulin secretion and insulin AUC.

Fasting insulin sensitivity was calculated as QUICKI index [38] and insulin sensitivity during the OGTT as the OGIS index [28].

The total glucose, insulin, and C-peptide excursions during the OGTT were calculated as the glucose AUC using the trapezoidal rule.

### 2.3. Bias and Study Size

Information bias was mitigated in this study as every PwCF should undergo an annual OGTT screening to early detect glucose intolerance. While it is possible that more severe patients may have been screened more intensively, we specifically excluded OGTT performed in acutely ill patients, according to the inclusion criteria. On the other hand, it is not advisable to conduct an OGTT on diabetic patients, so patients that resulted diabetic after measuring fasting glycemia were not allowed to continue the test. Moreover, the test is less tolerated by young patients, and they may have not completed the test at a higher rate than older patients. Survival bias was a possible source of selection bias that was tested, analyzing the raw trend lines of each outcome variable.

The number of patients tested in the recruiting centers during the study period determined the sample size.

### 2.4. Quantitative Variables and Statistical Methods

Age was binned with 5-year bins from 10 to 35 years, and patients outside this range were grouped in two extreme groups due to small group sizes.

Most continuous variables were not Gaussian-distributed, and all are reported as median (50th percentile) and interquartile range (IQR; 25th and 75th percentiles). Discrete variables are reported as the number and proportion of subjects with the characteristic of interest.

The relationship between age groups and outcomes was described through quantile regression to estimate within each age group the median, the lower, and the upper quartile of each outcome, adjusting both for sex and pancreatic insufficiency status.

### 2.5. Italian Reference Values

To limit the influence of outliers, all continuous variables besides age were winsorized using a tail of 0.01, meaning that values under the 1st percentile were put equal to the 1st percentiles, and values above the 99th percentile were put equal to the 99th percentile [39].

The 25th, 50th and 75th percentiles of each OGTT-related variable were estimated from a quantile regression model employing the variable as outcome and age (continuous, years), sex (discrete, 0 = female; 1 = male) and pancreatic insufficiency (0 = no; 1 = yes) as predictors [40]. Because of missing data for most variables, we fitted quantile regression using multiple imputation by chained equation (MICE), as detailed in the Appendix A.

## 3. Results

### 3.1. Participants Characteristics

A total of 369 patients were included in statistical analysis, their characteristics are presented in Table 1. Median (IQR) patient age was 19 (15, 24) years), ranging from 6 to 56 years, and 56% of patients were females. The most frequent CFTR mutation was the F508del (23% were F508del homozygous, and 43% were F508del heterozygous), and 79% of the patients were pancreatic insufficient. While 90% of patients were of normal weight, only 42% were above the BMI target set by the Cystic Fibrosis Foundation. Glucose tolerance was normal in 65% of the patients, while 8.1% resulted affected by CFRD without fasting hyperglycaemia (patients affected by CFRD at basal measurement were not allowed to continue the OGTT).

### 3.2. Main Results

Table 2 shows patient age distribution and count by age group, stratified by sex and pancreatic insufficiency, while the relationships between outcomes and age groups are shown in Figure 1, Figure 2 and Figure 3.

Visual inspection of Figure 1 shows increasing glucose values in the second hour of the OGTT, and decreasing insulin and C-peptide values, going from younger to older ages (see Panel A and B). This was paired with decreasing β cell glucose sensitivity and insulin secretion (see Panel C), increasing insulin clearance (see panel D) and greater fasting insulin sensitivity (see Panel E). Despite these overall trends, most variables showed a deviation from linearity at one or both extreme age groups (≤10 and >35 years). Indeed, contrary to the overall trend highlighted in Figure 1, secretory parameters (C-peptide at all time points, β cell glucose sensitivity, basal and total insulin secretion) of patients ≤10 were significantly lower than in patients (10,15] years old (see Table 3). On the other hand, patients >35 years of age showed similar, if not improved (see C-peptide and β cell glucose sensitivity in Table 3) than younger patients, instead of continuing the deteriorating trend.

Outcomes stratified by pancreatic status showed the greatest differences between groups when compared to differences between sexes (see Figure 2), both in glucose tolerance and insulin secretory parameters (see Figure 3).

### 3.3. Italian Reference Values

To produce reference values of the OGTT parameter, patient selection was performed to obtain a sample that was of uniform composition and sufficiently sized on the age scale. We did not have a sufficiently sized sample to flexibly model the deviation occurring at extreme age groups (≤10 and >35) and thus, the analysis was limited to central age groups as they showed a more linear trend. We chose to limit the sample size to post-puberal patients (≥13 years old for females and ≥15 years for males) for three reasons: (1) exploratory analysis identified a peak in secretory parameters in the age group that includes puberty; (2) an increase in secretory parameters occurring with puberty seems compatible with other changes occurring during puberty (i.e. greater insulin resistance [41]); and (3) the sample was not sufficiently sized before puberty to model a slope change and the sample generally exhibited a linear trend for all variables, excluding patient before puberty (Appendix A show linear trends of OGTT-related parameters stratified by sex and puberty status). Patients >35 years old were also excluded from the analysis for similar reasons: (1) visual inspection, and Table 3 showed at least no deterioration, and in the case of β cell glucose sensitivity, an improvement of insulin secretory parameters after the 30–35 years age group; (2) these changes are compatible with a survival bias; and (3) the sample was not sufficiently sized to model age groups >35 years.

Figure 4 shows point estimates from quantile regression of all OGTT-related variables, stratified by sex. Most variables show an either increasing or decreasing trend with age: glucose tolerance (see raw glucose values in the second hour of the OGTT and glucose AUC) degrades with age, but it does not seem related to increasing insulin resistance, as insulin sensitivity (both fasting as QUICKI or during OGTT as OGIS) is stable throughout the age range; insulin secretion (expressed by raw insulin and C-peptide values, and modeled basal and total insulin secretion) generally degrades with age, accordingly with decreasing β cell glucose sensitivity and increasing insulin clearance. We deem all changes going from puberty to 35 years of age clinically significant, except for fasting glucose, glucose values in the first hour of OGTT, and insulin sensitivity parameters, that have shown minimal changes. Appendix A show point estimates and 95% confidence intervals from quantile regression of all OGTT-related variables stratified by sex, while Appendix A shows β-cell glucose sensitivity stratified by age groups and glucose tolerance categories.

## 4. Discussion

In this study we present a description of the distribution and Italian reference quartiles for OGTT parameters and their AUCs, β cell function, insulin clearance, and insulin sensitivity. For all variables, we described the distribution from pre-puberal age to old adults accounting for differences in sex and pancreatic insufficiency between age groups, while we produced reference values for post-puberal patients and young adults that are age- and sex-specifically adjusted for pancreatic insufficiency. Our data confirmed an approximately linear degradation of glucose tolerance and insulin secretion during adulthood of PwCF and up until 35 years of age, with trend deviations occurring both in younger and older patients (these age groups were therefore excluded from the modeling of reference quartiles). We provide suggestive findings in the two extreme age groups: our data and the comparison with fasting insulin reference data for the general population (see Appendix A) suggest that a peak in insulin secretion occurs approximately across puberty, both in PwCF and in the general population. On the other hand, patients >35 years of age show better glucose tolerance and insulin secretion than younger peers, seemingly identifying survivors with preserved pancreatic function.

*Puberty*. Puberty seems associated with an increase in insulin secretory parameters. Our exploratory analysis highlighted a peak in C-peptide values, and modeled β cell glucose sensitivity, basal and total insulin secretion. The dynamics of insulin secretion across puberty are poorly studied even in the general population, although available evidence seems to confirm our data. Insulin secretion in puberty was studied cross-sectionally by [42] in 23 subjects using the hyperglycemic clamp technique; they found that adolescents display greater insulin and C-peptide responses when compared to both pre-adolescents and adults, even though glucose responses were similar for all groups. The authors of [43] also studied insulin secretory capacity using the hyperglycemic clamp in 133 subjects cross-sectionally, and 24–27 subjects longitudinally, both analyses showing an increase of insulin secretory capacity across puberty and a decrease later in adulthood. Moreover, combined general-population reference data for fasting insulin available in the Appendix A [44,45,46,47] show an increasing trend before puberty and a decreasing trend thereafter. Superimposing cystic fibrosis quartiles of fasting insulin produced in this study to combine reference quartiles for the general population shows that at least fasting insulin, that is reflection of insulin secretion, seems generally preserved in young PwCF, although a faster degeneration with age is evident in PwCF, as expected.

*Older patients*. PwCF >35 years old show a better glucose tolerance and insulin secretion than younger patients. We provide two possible explanations for this phenomenon: (1) we only included patients without CFRD, as OGTT is not feasible in those patients. As CFRD prevalence increases with age [26], and it represents the final stage of progressive glucose tolerance and insulin secretion degradation, patients with lower glucose tolerance and lower insulin secretion are progressively excluded in our analysis (selection bias); and (2) as CFRD is associated with lower survival, PwCF that live longer tend to show better pancreatic function (survival bias). In this cross-sectional analysis, patients with better pancreatic function are present in all age groups, but the combination of the aforementioned factors cause a deviation from linearity in the extreme age group of patients >35 years old; therefore, they were excluded from the analysis. Older CFRD-free PwCF may represent a model to better understand the mechanism behind CFRD occurrence.

*References values in post-puberal patients*. We previously published OGTT-related variable quartiles from a smaller sample recruited at a single center [27]. In this study, we can confirm the relationship between OGTT-variables and age while introducing novel findings: (1) glucose tolerance deteriorates with age seemingly starting after puberty, in particular in the second hour of the OGTT; (2) glucose tolerance derangements are seemingly due to reduced insulin secretion, as highlighted by reduced raw insulin values, both fasting and particularly during OGTT, but perhaps more importantly as highlighted by a reduction of the β cell response to an increase in glucose concentration that delay the insulin response, causing glucose elevation in the second half of the OGTT; (3) insulin sensitivity does not deteriorate with age, reinforcing the hypothesis that glucose derangements recorded in PwCF are caused by an insulin secretory defect and not by an increased insulin resistance; (4) contributing to insulin secretory defects, both basal and OGTT insulin clearance appear to increase with age, although this may be in fact due to lower circulating and secreted insulin levels that do not saturate the systemic (mostly hepatic and renal) abilities to clear insulin; and (5) β cell dysfunction highlighted by reduced β cell glucose sensitivity is closely related to pancreatic dysfunction.

*Sex differences*. In comparison with [27], sex-related differences shown in this study seem reduced. Here we accounted for differences in pancreatic insufficiency, and, perhaps more importantly, we included only post-puberal patients, as younger patients displayed a puberty-related increment in insulin secretion that we could not model reliably. It is plausible that the inclusion of pre-puberal patients in our previous work displaced the overall linear trends of studied parameters. In contrast, when accounting for age differences in puberty incidence, the two sexes show similar trends.

*Pancreatic insufficiency*. Pancreatic sufficiency status was the greatest determinant of glucose tolerance and insulin secretion. There is a known correlation and plausible biological mechanism linking pancreatic insufficiency and β cell function, with pancreatic insufficient patients showing lower glucose tolerance and lower insulin secretion [48,49]. In PwCF with pancreatic insufficiency, exocrine pancreas is replaced in large amounts by fibrotic and/or fatty tissue, while islet mass is relatively conserved [50]. Emerging theories have suggested that a crosstalk between pancreatic ductal epithelial cells and beta cells may be a main contributor to beta cell dysfunction in PwCF, as CFTR is expressed in ducts and diseased ducts may influence β cell function through exocrine-derived proinflammatory factors [51]. In this study, we included pancreatic sufficiency status as a covariate to adjust for difference in pancreatic insufficiency prevalence in calculated quartiles, but we do not provide separate references for pancreatic sufficient and insufficient patients, as outcomes related to glucose tolerance and insulin secretion are likely to occur independently from pancreatic insufficiency when considering the endocrine pancreatic function.

The study has some limitations. As has been noted, the sample size and age distribution prevented to acceptably model patients in the extreme age groups, and so they were excluded from the analysis. No gold-standard methods were used to measure insulin secretion and sensitivity, although we consider these reasonable constraints of a relatively large study. A potential limitation of the modeling of β cell function employed in this study is the use of the C-peptide kinetic model by [52], as previously reported [27]. On the other hand, this study improved on the previous work [27] by avoiding the trajectories bias produced by including puberal and pre-puberal patients in the analysis, by improving sample size with greater recruitment of patients from multiple sites, and finally, by including the contribution of pancreatic sufficiency status in the developed quantiles.

In conclusion, we provide a description of the distribution and Italian reference quartiles for OGTT-related variables. We have already shown how these parameters are likely to predict overt CFRD [19], how the deterioration of these parameters is linked with lung function [22] and reduced adult stature [25], and finally, how these parameters are seemingly unaffected by the lumacaftor/ivacaftor combination therapy when administered in post-puberal PwCF [11]. These references provide a necessary tool to not only identify PwCF at risk for CFRD or other cystic fibrosis-related complications, but also to evaluate the effects of promising pharmacological therapies. As administration in early ages of new therapies has the greatest potential to provide significant improvements in glucose tolerance and insulin secretion, a better characterization of the natural history of these parameters during puberty is strongly needed.

## Figures and Tables

**Figure 1 jpm-13-00469-f001:**
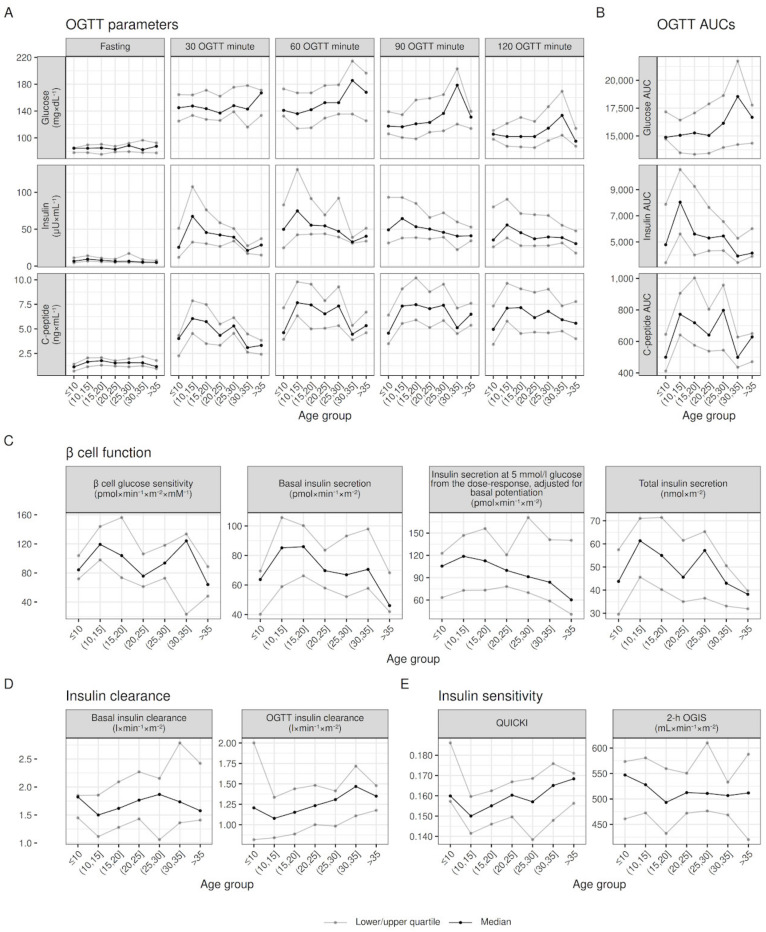
Within the age-group raw distribution of OGTT parameters, β cell function, insulin clearance, and insulin sensitivity. Quartiles values adjusted for sex and pancreatic insufficiency to account for differences in their prevalence between age groups; lines joining quartiles between age groups do not represent modeling for age.

**Figure 2 jpm-13-00469-f002:**
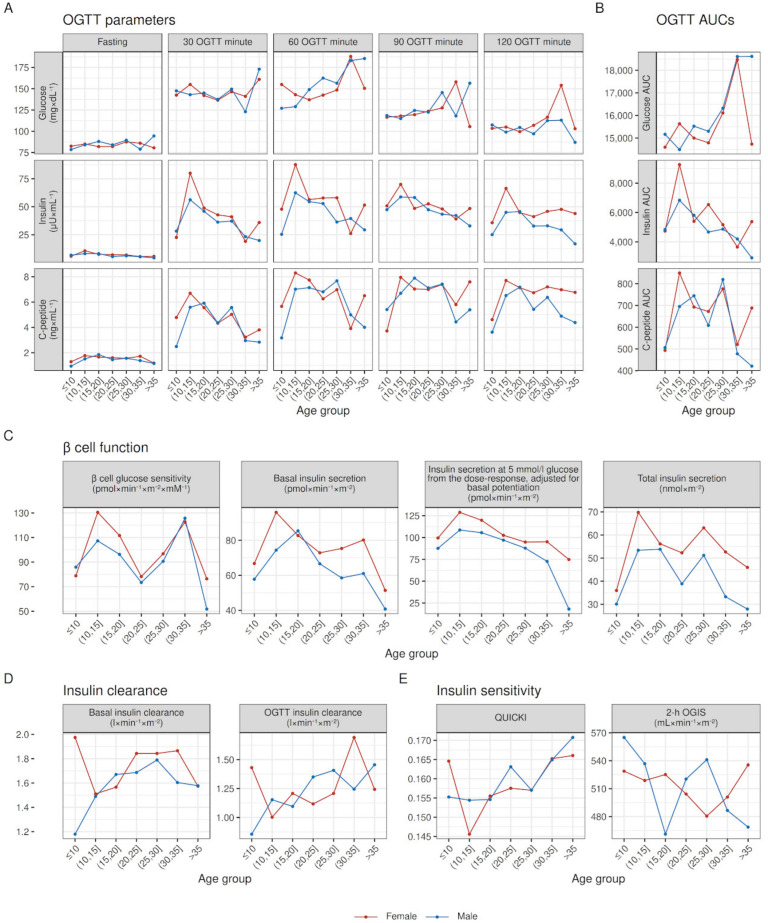
Within age-group raw medians of OGTT parameters, β cell function, insulin clearance, and insulin sensitivity, stratified by sex. Median values adjusted for pancreatic insufficiency to account for differences in pancreatic insufficiency prevalence between age groups; lines joining medians between age groups do not represent modeling for age.

**Figure 3 jpm-13-00469-f003:**
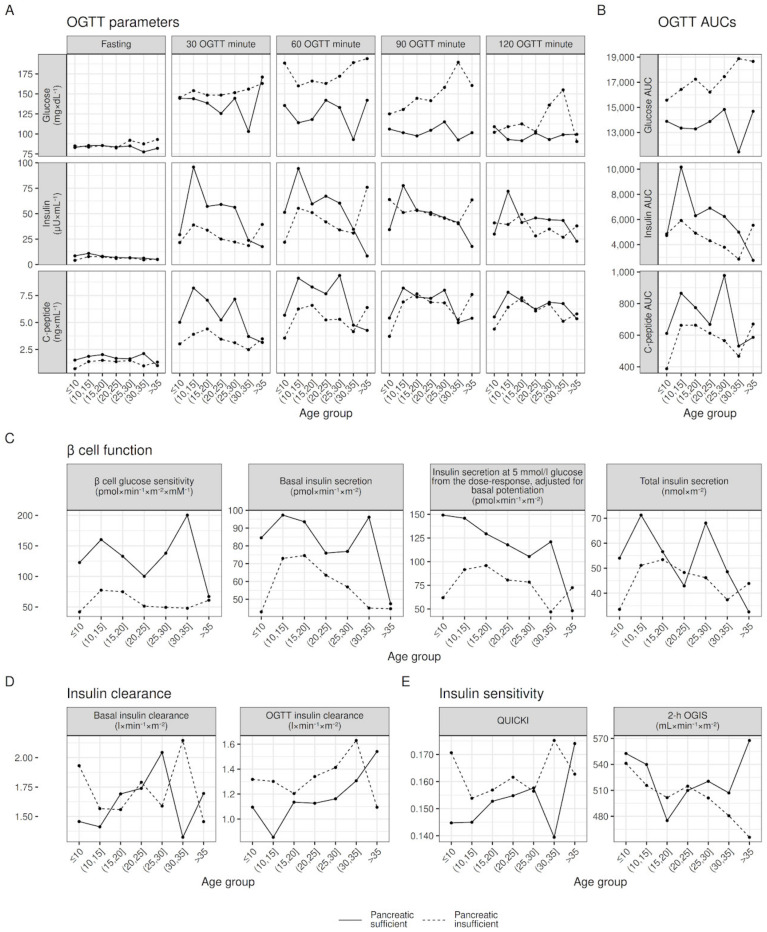
Within age-group raw medians of OGTT parameters, β cell function, insulin clearance, and insulin sensitivity, stratified by pancreatic insufficiency. Median values adjusted for sex insufficiency to account for differences in sex between age groups; lines joining medians between age groups do not represent modeling for age.

**Figure 4 jpm-13-00469-f004:**
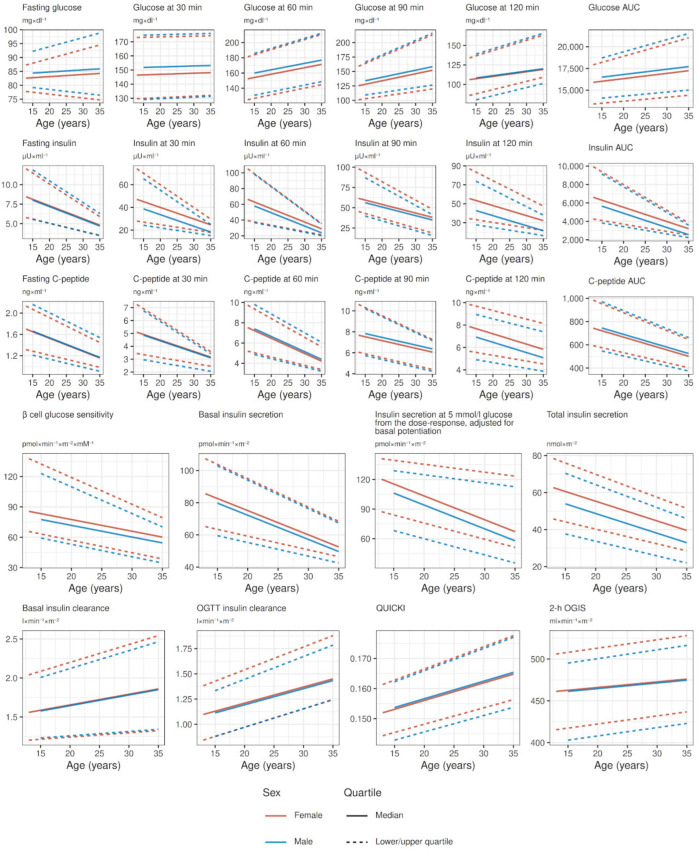
Reference values of OGTT-related variables for both sexes in the age range between puberal age (>13 years for females, >15 years for males) and 35 years, derived from point estimates of quantile regression.

**Table 1 jpm-13-00469-t001:** Patient characteristics.

Characteristic	N = 369 ^1^
Age (years)	19 (15, 24)
Sex	
*Female*	56%
*Male*	44%
CFTR gene mutation	
*F508del homozygous*	23%
*F508del heterozygous*	43%
*Other*	34%
Pancreatic insufficiency	79%
Liver disease	25%
Pseudomonas aeruginosa infection	65%
Burkholderia cepacia infection	4.1%
Liver transplant	0.3%
Lung transplant	0.3%
CFTR modulator therapy	
*No therapy*	88.9%
*Ivacaftor*	2.9%
*Tezacaftor/Ivacaftor*	0.3%
*Lumacaftor/Ivacaftor*	7.9%
Weight (kg)	54 (46, 63)
Weight z-score (CDC growth charts)	−0.01 (−0.03, 0.02)
Height (cm)	161 (154, 169)
Height z-score (CDC growth charts)	−0.0008 (−0.0020, 0.0003)
BMI (kg × m^−2^)	20.5 (18.7, 22.8)
BMI z-score (CDC growth charts)	−0.003 (−0.016, 0.010)
BMI category	
*Underweight*	3.6%
*Normal weight*	90%
*Overweight*	4.9%
*Obese*	1.1%
BMI Cystic Fibrosis Foundation recommendations	
*Below target*	58%
*Above target*	42%
FEV1 (% of predicted)	88 (70, 103)
FVC (% of predicted)	98 (84, 108)
C-reactive protein (mg × dL^−1^)	0.25 (0.09, 0.99)
HbA1C (%)	5.44 (5.20, 5.90)
Glucose tolerance category	
*Normal glucose tolerance*	65%
*Normal glucose tolerance with impaired fasting glucose*	0.6%
*Indeterminate glucose tolerance*	12%
*Impaired glucose tolerance*	15%
*Cystic fibrosis-related diabetes without fasting hyperglycemia*	8.1%
*Cystic fibrosis-related diabetes with fasting hyperglycemia*	0%

^1^ Median (IQR).

**Table 2 jpm-13-00469-t002:** Patient count and age distribution by age group.

		Age Group (Range in Years)						
Characteristic	Overall, N = 369 ^1^	≤10, N = 12 ^1^	(10,15], N = 92 ^1^	(15,20], N = 101 ^1^	(20,25], N = 80 ^1^	(25,30], N = 41 ^1^	(30,35], N = 25 ^1^	>35, N = 18 ^1^
Age (years)	19 (15, 24)	9 (9, 10)	13 (11, 14)	17 (16, 19)	22 (21, 24)	27 (26, 28)	32 (31, 33)	43 (37, 46)
Sex								
Female	206 (56%)	8 (67%)	49 (53%)	47 (47%)	51 (64%)	25 (61%)	15 (60%)	11 (61%)
Male	163 (44%)	4 (33%)	43 (47%)	54 (53%)	29 (36%)	16 (39%)	10 (40%)	7 (39%)
Pancreatic status								
Pancreatic sufficient	77 (21%)	4 (33%)	18 (20%)	24 (24%)	13 (16%)	9 (22%)	4 (16%)	5 (28%)
Pancreatic insufficient	292 (79%)	8 (67%)	74 (80%)	77 (76%)	67 (84%)	32 (78%)	21 (84%)	13 (72%)

^1^ Median (IQR); N (%).

**Table 3 jpm-13-00469-t003:** Differences between patients in the extreme and adjacent age groups, controlling for differences in sex and pancreatic insufficiency status.

	(10,15] Years vs. ≤10 Years			>35 Years vs. (30,35] Years		
Characteristic	Difference	95% CI	*p*-Value	Difference	95% CI	*p*-Value
Fasting glucose (mg × dL^−1^)	2.0	−4.0, 8.0	0.5	1.0	−20, 22	0.9
Glucose (mg × dL^−1^) 30 OGTT minute	0.00	−12, 12	0.9	17	−18, 52	0.3
Glucose (mg × dL^−1^) 60 OGTT minute	−9.0	−33, 15	0.5	5.0	−43, 53	0.8
Glucose (mg × dL^−1^) 90 OGTT minute	−1.0	−15, 13	0.9	−19	−71, 33	0.5
Glucose (mg × dL^−1^) 120 OGTT minute	−8.0	−20, 4.0	0.2	−39	−85, 7.3	0.10
Fasting insulin (μu × mL^−1^)	2.3	−0.34, 5.0	0.087	−0.05	−3.3, 3.2	0.9
Insulin (μU × mL^−1^) 30 OGTT minute	30	−36, 96	0.4	10	−8.7, 29	0.3
Insulin (μU × mL^−1^) 60 OGTT minute	11	−19, 42	0.5	13	−6.0, 31	0.2
Insulin (μU × mL^−1^) 90 OGTT minute	−9.1	−68, 50	0.8	7.7	−27, 42	0.7
Insulin (μU × mL^−1^) 120 OGTT minute	5.2	−21, 31	0.7	−2.9	−32, 26	0.8
Fasting C−peptide (ng × mL^−1^)	0.48	0.20, 0.76	<0.001	−0.01	−0.60, 0.58	0.9
C-peptide (ng × mL^−1^) 30 OGTT minute	1.3	0.20, 2.4	0.021	0.65	−0.08, 1.4	0.078
C-peptide (ng × mL^−1^) 60 OGTT minute	3.2	2.1, 4.3	<0.001	1.6	0.23, 2.9	0.023
C-peptide (ng × mL^−1^) 90 OGTT minute	3.5	1.7, 5.2	<0.001	1.8	−0.40, 4.0	0.10
C-peptide (ng × mL^−1^) 120 OGTT minute	2.8	1.7, 3.8	<0.001	−0.30	−3.3, 2.7	0.8
β cell glucose sensitivity (pmol × min^−1^ × m^−2^ × mM^−1^)	30	13, 46	<0.001	19	0.96, 37	0.040
Basal insulin secretion (pmol × min^−1^ × m^−2^)	21	5.4, 37	0.009	−5.0	−32, 22	0.7
Total insulin secretion (nmol × m^−2^)	25	15, 35	<0.001	−3.6	−18, 11	0.6
Insulin secretion at 5 mmol/L glucose from the dose-response, adjusted for basal potentiation (pmol × min^−1^ × m^−2^)	19	−6.7, 45	0.14	18	−40, 77	0.5
Basal insulin clearance (L × min^−1^ × m^−2^)	−0.24	−0.54, 0.06	0.12	−0.26	−1.2, 0.69	0.6
OGTT insulin clearance (L × min^−1^ × m^−2^)	−0.29	−1.1, 0.52	0.5	−0.29	−0.86, 0.28	0.3
Glucose AUC	7.5	−1330, 1345	0.9	270	−3897, 4437	0.9
Insulin AUC	864	−2402, 4130	0.6	1640	−202, 3482	0.078
C-peptide AUC	290	195, 384	<0.001	133	−26, 292	0.10
QUICKI	−0.01	−0.02, 0.00	0.15	0.00	−0.02, 0.01	0.9
2−h OGIS (mL × min^−1^ × m^−2^)	−16	−73, 40	0.6	−0.23	−83, 82	0.9

## Data Availability

Data available upon reasonable request due to privacy restrictions.

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
