# Peer review of "Distribution of OGTT-Related Variables in Patients with Cystic Fibrosis from Puberty to Adulthood: An Italian Multicenter Study"

_jpm, 2023, doi:10.3390/jpm13030469_

Round 1

Reviewer 1 Report

This work is an extension of a previous study [1]. Similar to the previous study, the authors perform an oral glucose tolerance test (OGTT) on Italian people with cystic fibrosis, collect information on OGTT-related outcomes (glucose, insulin, C-peptide) as well as other clinical data, and use quantile regression to investigate the association between outcomes and clinical characteristics. This work is similar to the previous study [1], although the authors point out that it is an improvement since the current work  includes puberal and pre-puberal patients in the analysis; recruits patients from a larger number of sites (3 CF centres rather than 1) and considers the association between pancreatic sufficiency and outcomes.

The authors state that the “aim of this study is to describe at the Italian level the progression of OGTT parameters [and other outcomes] in order to understand their natural history, provide national references and as a tool to evaluate the effects of new therapies on glucose tolerance mechanisms. ” It is not clear to me what is meant by “describe at the Italian level the progression of OGTT parameters” and I am unsure how the data presented can be used to achieve the three purposes stated.  

·         This is a cross-sectional study where data on individuals were collected during one day. To understand the natural history, I would typically expect to see a longitudinal study, where data are collected at several time points over a longer period of time, for example in [2].

·         A total of 369 patients were included in the statistical analysis. The authors state that the sample size was determined by the “number of patients tested in the recruiting centres during the study period”, but there is no justification or discussion as to whether this is large enough for results to provide reliable national reference values.

·         The authors conclude that the quantile references provide a tool to evaluate the effects of pharmacological therapies (CFTR  modulators). Could this be explained? I imagine one could compare OGTT outcomes among groups of people taking CFTR modulators to that expected if they did not initiate treatment with modulators (i.e. the reference values presented here), but the group of people taking CFTR modulators may be very different to the sample used to create reference values in terms of confounding characteristics. Estimating the effect of CFTR modulators on health outcomes would require a formal causal inference approach as in [3]. This can be done without reference quantiles.

I think it would be useful if the authors provide clearer aims and objectives so that it is easier to assess whether these have been achieved in an appropriate way.

Finally, the provided supplementary files were the figures already given in the manuscript. I believe there was additional information on how multiple imputation was imputed that is missing.

[1] Battezzati A, Bedogni G, Zazzeron L, Mari A, Battezzati PM, Alicandro G, Bertoli S, Colombo C. Age- and Sex-Dependent Distribution of OGTT-Related Variables in a Population of Cystic Fibrosis Patients. J Clin Endocrinol Metab. 2015 Aug;100(8):2963-71. doi: 10.1210/jc.2015-1512. Epub 2015 Jun 9. PMID: 26057180.

[2] Sterescu AE, Rhodes B, Jackson R, Dupuis A, Hanna A, Wilson DC, Tullis E, Pencharz PB. Natural history of glucose intolerance in patients with cystic fibrosis: ten-year prospective observation program. J Pediatr. 2010 Apr;156(4):613-7. doi: 10.1016/j.jpeds.2009.10.019. Epub 2009 Dec 3. PMID: 19962154.

[3] Newsome SJ, Daniel RM, Carr SB, Bilton D, Keogh RH. Using Negative Control Outcomes and Difference-in-Differences Analysis to Estimate Treatment Effects in an Entirely Treated Cohort: The Effect of Ivacaftor in Cystic Fibrosis. Am J Epidemiol. 2022 Feb 19;191(3):505-515. doi: 10.1093/aje/kwab263. PMID: 34753177; PMCID: PMC8914944.

Author Response

Reviewer 1:

  • This is a cross-sectional study where data on individuals were collected during one day. To understand the natural history, I would typically expect to see a longitudinal study, where data are collected at several time points over a longer period of time, for example in [Sterescu AE, Rhodes B, Jackson R, Dupuis A, Hanna A, Wilson DC, Tullis E, Pencharz PB. Natural history of glucose intolerance in patients with cystic fibrosis: ten-year prospective observation program. J Pediatr. 2010 Apr;156(4):613-7. doi: 10.1016/j.jpeds.2009.10.019. Epub 2009 Dec 3. PMID: 19962154.].
    • We agree, we have renamed the paper as “Distribution of OGTT-related variables in patients with Cystic Fibrosis from puberty to adulthood: an Italian multicenter study” and corrected the manuscript text accordingly.
  • A total of 369 patients were included in the statistical analysis. The authors state that the sample size was determined by the “number of patients tested in the recruiting centres during the study period”, but there is no justification or discussion as to whether this is large enough for results to provide reliable national reference values.
    • We recruited all patients referring to the clinical centres involved in the study during the study period. As we did not have an inferential aim, traditional sample size calculation was not performed. Indeed we recognize the limit imposed by our sample size and so we did not use the statistical method traditionally employed in growth charts (e.g. the LMS method) that assumed a priori that adequate sample size is met. We used quantile regression precisely because it provides a margin of errors and confidence intervals that give for each quantile, outcome and population sub-strata quantification of uncertainty.
  • The authors conclude that the quantile references provide a tool to evaluate the effects of pharmacological therapies (CFTR modulators). Could this be explained? I imagine one could compare OGTT outcomes among groups of people taking CFTR modulators to that expected if they did not initiate treatment with modulators (i.e. the reference values presented here), but the group of people taking CFTR modulators may be very different to the sample used to create reference values in terms of confounding characteristics. Estimating the effect of CFTR modulators on health outcomes would require a formal causal inference approach as in [Newsome SJ, Daniel RM, Carr SB, Bilton D, Keogh RH. Using Negative Control Outcomes and Difference-in-Differences Analysis to Estimate Treatment Effects in an Entirely Treated Cohort: The Effect of Ivacaftor in Cystic Fibrosis. Am J Epidemiol. 2022 Feb 19;191(3):505-515. doi: 10.1093/aje/kwab263. PMID: 34753177; PMCID: PMC8914944.]. This can be done without reference quantiles.
    • Quartile reference will provide a tool to detect differences in glucose tolerance and insulin secretory parameters. We agree that once the existence of differences will be ascertained, an exploration of changes of confounding characteristics and formal causal inference studies will be necessary. We think that reference data in the pre-modulator era will provide valuable insight in the near future when virtually no patient will be modulator naive, at least within specific mutations. Moreover our investigation highlighted trends of insulin secretion related to age that are not well understood even in the general population (see elevation of insulin secretion across puberty) that needs to be taken into account when evaluating the effects of treatments on the affected parameters.
  • The authors state that the “aim of this study is to describe at the Italian level the progression of OGTT parameters [and other outcomes] in order to understand their natural history, provide national references and as a tool to evaluate the effects of new therapies on glucose tolerance mechanisms. ” It is not clear to me what is meant by “describe at the Italian level the progression of OGTT parameters” and I am unsure how the data presented can be used to achieve the three purposes stated.. I think it would be useful if the authors provide clearer aims and objectives so that it is easier to assess whether these have been achieved in an appropriate way.
    • Agreed, we re-stated our aims in order to be more adherent to the conclusions that have been supported by our data.
  • Finally, the provided supplementary files were the figures already given in the manuscript. I believe there was additional information on how multiple imputation was imputed that is missing
    • Indeed, the appendix was not included in the supplementary data, we have now uploaded it with the revised version of the manuscript.

Reviewer 2 Report

The authors present a natural history study outlining progression of dysglycemia in an Italian cohort. They also implement statistical modeling to describe beta cell dysfunction by utilizing modified OGTT techniques that include more frequent time points than a standard OGTT.  The contribution of the study is by providing a strong longitudinal description of glycemic progression in this population.  In addition, they provide pathophysiologic evidence regarding beta cell dysfunction.  The authors provide a strong discussion and overall synthesis of the data. I’d recommend further discussion on characterization of indeterminate glycemia, specifically discussing whether the current cut-off is sufficient for the population being studied (other studies have suggested a lower cutoff for INDET may be needed based on beta cell modeling). Graphical representation of OGTT data across age groups and variables are also well done.

Author Response

Reviewer 2:

  • I’d recommend further discussion on characterization of indeterminate glycemia, specifically discussing whether the current cut-off is sufficient for the population being studied (other studies have suggested a lower cutoff for INDET may be needed based on beta cell modeling).
    • We thank the reviewer for the suggestion. We have compared β cell glucose sensitivity across glucose tolerance category and age group, performing pairwise comparison of the “Indeterminate glucose tolerance” group with other groups. We see that the indeterminate group tends to have similar β cell glucose sensitivity to the “Normal glucose tolerance” group only in young age groups (<20 years old), while it seems similar to more severe glucose tolerance groups in later stages. The numerosity of glucose tolerance groups across age groups may have affected these results, but in the paper we show that β cell glucose sensitivity has a dynamic relationship with age, seemingly showing a peak across puberty, and prevalence of indeterminate glycemia is also higher across puberty. We think that further studies are required to further discern the contribution of age in the characterization of endocrine pancreatic function across glucose tolerance categories. We have included this analysis in the appendix